# Impact of Chemical Endocrine Disruptors and Hormone Modulators on the Endocrine System

**DOI:** 10.3390/ijms23105710

**Published:** 2022-05-20

**Authors:** Valentina Guarnotta, Roberta Amodei, Francesco Frasca, Antonio Aversa, Carla Giordano

**Affiliations:** 1Department of Health Promotion, Mother and Child Care, Internal Medicine and Medical Specialties “G. D’Alessandro” (PROMISE), Section of Endocrinology, University of Palermo, Piazza delle Cliniche 2, 90127 Palermo, Italy; valentina.guarnotta@unipa.it (V.G.); roberta.amodei@gmail.com (R.A.); 2Endocrinology Section, Department of Clinical and Experimental Medicine, Garibaldi Nesima Hospital, University of Catania, 95122 Catania, Italy; frascafranco@gmail.com; 3Department of Experimental and Clinical Medicine, Section of Endocrinology, Magna Graecia University of Catanzaro, 88100 Catanzaro, Italy; aversa@unicz.it

**Keywords:** bisphenol, physical agents, phthalates, adrenal, thyroid, parathyroid, beta-cell, ovary, testis

## Abstract

There is growing concern regarding the health and safety issues of endocrine-disrupting chemicals (EDCs). Long-term exposure to EDCs has alarming adverse health effects through both hormone-direct and hormone-indirect pathways. Non-chemical agents, including physical agents such as artificial light, radiation, temperature, and stress exposure, are currently poorly investigated, even though they can seriously affect the endocrine system, by modulation of hormonal action. Several mechanisms have been suggested to explain the interference of EDCs with hormonal activity. However, difficulty in quantifying the exposure, low standardization of studies, and the presence of confounding factors do not allow the establishment of a causal relationship between endocrine disorders and exposure to specific toxic agents. In this review, we focus on recent findings on the effects of EDCs and hormone system modulators on the endocrine system, including the thyroid, parathyroid glands, adrenal steroidogenesis, beta-cell function, and male and female reproductive function.

## 1. Introduction

Endocrine disruptors are exogenous agents that interfere with endocrine actions, having a deleterious effect on them and showing a direct cause–effect relationship in exposed subjects, offspring, or subpopulations [1,2]. Fetal exposure to endocrine disruptors occurs through the placenta [3] and breast feeding [4], while adults are exposed through food, inhalation, and skin contact [5].

Endocrine disruptors may be derived from natural animal, human, or plant (phytoestrogen) sources. However, the best-known are chemical endocrine disruptors (EDCs). EDCs include industrial ones (dioxins, polychlorinated biphenyls (PCBs) and alkyphenols), agricultural ones (pesticides, herbicides, fungicides, and insecticides), phthalates, bisphenol A (BPA), drugs (mitotane, ketoconazole, cardiac glycosides, nitrofurans, carbamazepine, and astazene), and heavy metals [6]. Non-chemical compounds are generally represented by artificial light, radiation, temperature, and stress and can affect the endocrine system modulating hormonal functions [7,8,9]. EDCs mimic endocrine action by binding to many hormone receptors of different endocrine glands acting as agonists or antagonists [10]. Generally, there are two vehicles of interaction: membrane-bound receptors and nuclear receptors [11]. EDCs that bind membrane receptors can disrupt non-genomic signaling pathways.

However, endocrine disruptors should not be confused with endocrine modulators, which are compounds interacting with hormonal systems, without disrupting them. Endocrine modulators are generally quite benign to humans, even though it depends on the level in the finished product and the frequency of use. For example, many of us consume coffee, chocolate, or soy, and yet these substances are well known to interact with the hormonal system. However, if consumed in the conventional amounts they are harmless for humans. 

Today, EDCs are estimated to number above 4000, and there is increased pollution from these chemicals. Consequently, human health through known or unknown effects of these chemicals on hormonal systems is seriously involved. The principal cause able to explain the impact of endocrine-disrupting chemicals on human health is considered to be connected to the very high production and use of industrial and agricultural chemicals and their capacity to influence endocrine function. Primarily, endocrine disruptors may interfere with natural hormone synthesis, secretion, metabolism binding, elimination, and transport [11]. EDCs can impact different hormonal targets including hormone production and hormone receptor expression, acting as receptor agonists or antagonists. For instance, BPA has been reported to act as an agonist when it binds to estrogen receptors (ERs) and an antagonist binding to androgen receptors/an androgen receptor (AR). In addition, there are chemical compounds that act as hormone modulators having an impact on signaling pathways.

EDCs can affect the hormonal balance and result in developmental, reproductive, and behavioral abnormalities [12,13]. Recent studies showed a link between EDC exposure with obesity, metabolic syndrome, and type 2 diabetes [12,14].

The National Health and Nutrition Examination Survey (NHANES) database cross-sectionally analyzed 1721 adults (see NHANES and Observational Studies) reporting a positive association between diabetes and serum levels of 19 different persistent pollutants (including organochlorine pesticides, OCPs) [15,16,17,18,19].

The aim of this review is to focus on current knowledge of EDCs and chemical and non-chemical endocrine modulators on the endocrine system. We carefully evaluated the effects on the thyroid, parathyroid glands, adrenal glands, beta-cells, testes, and ovaries. In this paper, the main chemical and non-chemical compounds that may contribute to metabolic disruption or hormonal modulation will be discussed on the basis of their demonstrated or suggested action mechanisms.

## 2. Thyroid and Parathyroid Glands

Thyroid hormones are important for both development and metabolism in adult vertebrates. Hence, thyroid function and thyroid hormone action are highly regulated during both fetal life and adulthood. Proper thyroid function is dependent on physiological environmental factors such as iodine, selenium, and iron. However, in food and the environment there are some natural goiter-inducing agents that interfere with thyroid function including thiocyanates and isoflavones, nitrates, chlorates, and perchlorates. Nowadays, to the list of these natural thyroid interfering agents, several industrial chemicals, such as heavy metals, phthalates, and furans, which can interfere with thyroid function or thyroid hormone action, can be added.

Within the same individual, thyroid hormone levels range within a narrow interval and are strictly regulated by a genetic “set point”. By contrast, among different individuals within a given population the reference range may vary up to 10-fold compared to the individual range [20,21]. Hence, understanding these intra- and inter-individual variations is an important prerequisite to evaluate the effect of EDCs on human thyroid function. The thyroid function is regulated by a complex interplay between the hypothalamus (which releases the hormone TRH), the pituitary gland (which releases TSH), and the thyroid (which produces T4 and T3) [22]. The thyroid eminently secretes T4, which is converted into T3 by type 1 and 2 deiodinases (Dio1 and Dio2) [23]. The effect of thyroid-specific EDCs may occur at any level including the hypothalamus–pituitary–thyroid axis, thyroid hormone synthesis, release, transport, metabolism, and action on target tissues. The mechanism of interference is based mainly upon structural similarities between EDCs and thyroid hormones. These peculiar features of thyroid hormone production and metabolism make studies on EDCs very difficult [24] because EDC action may involve different targets and does not necessarily imply a detectable change in thyroid hormone levels. Hence, clinical manifestation of effects of EDCs on the thyroid may be very different compared to those of traditional thyroid diseases.

Several micronutrients are important in thyroid hormone production [25,26,27] and may also contribute to the effect of EDCs on thyroid hormone action. Iodine is crucial for thyroid hormone synthesis and is entrapped into thyrocytes by the sodium/iodide symporter (NIS) and several environmental chemicals may interfere with NIS function and iodine uptake, including perchlorate, chlorate, nitrate, and thiocyanate. Chlorate and nitrate may be present in water supplies, and thiocyanate in cigarette smoke and vegetables. Inhibition of iodide uptake by these anions may be important in iodine-deficient areas [28] and pregnant women [29]. Moreover, thiocyanates and isoflavones may inhibit the thyroperoxidase (TPO) enzyme, which is essential for thyroid hormone synthesis [30,31].

EDCs can also reduce circulating levels of thyroid hormones inducing the liver enzymes responsible for T4/T3 clearance [32,33]. Further, they can influence thyroid hormone binding to carrier proteins, such as phenolic compounds including PCBs and PBDEs [34,35,36,37,38,39]. EDCs such as halogenated biphenyls and biphenyl ethers, displaying a structure similar to thyroid hormone, have been demonstrated to impact the thyroid hormone metabolism, while others, such as the insecticide fipronil, have an impact on thyroid hormone transport [40].

The most widely studied EDCs having an effect on thyroid function include the ones that will now be listed.

Perchlorate. Experimental studies indicate that human exposure to a perchlorate level of about 5.0 mcg/kg/die is able to significantly reduce thyroid iodine uptake [41]. However, human toxicology studies suggest that only high doses of perchlorate are able to significantly inhibit thyroid hormone synthesis [41]. Perchlorate may significantly affect thyroid hormone action in newborn development [42], because infants are particularly vulnerable to thyroid hormone insufficiency [43] and perchlorate levels may be very high in breast milk [44]. In addition, a recent study found a relationship between exposure to perchlorate of pregnant women and reduced cognitive function in newborns [29]. 

Polychlorinated biphenyls. In contrast with animal models, data in the literature describing the relationship between PCB exposure and variations in thyroid hormone levels in humans are scanty and controversial. On the other hand, a considerable number of papers describe a correlation between fetal exposure to PCBs and a variety of cognitive defects in children [45]. Hence, several authors hypothesize that PCBs may act as thyroid hormone modulators on the signaling in the fetal nervous system.

Polybrominated diphenyl ethers. In a manner similar to PCBs, several studies indicate a significant negative correlation between cord blood polybrominated diphenyl ether (PBDEs) levels and cognitive function, with respect to full-scale, verbal, and performance IQ [46]. Although several studies were able to detect significant levels of PBDEs in the blood of pregnant women [47], in cord blood, and in breast milk [48,49], a direct effect on human thyroid function remains elusive [50,51,52,53] and they can be currently considered as hormone modulators.

Phthalates. Unlike PCBs and PBDEs, phthalates are able to significantly affect thyroid function in exposed subjects. Indeed, several studies have described a correlation between urinary levels of phthalates and thyroid function. In particular, some studies have described a negative association between urinary phthalates and serum-free and total T4 [47,54]. Moreover, other studies have indicated that urinary phthalates are positively associated with serum TSH [55], while another study found a positive correlation between phthalate intake and serum TSH in Taiwanese children [56]. 

Interestingly, phthalates may behave as both a thyroid receptor (TR) agonist and a TR antagonist [57].

Bisphenol A. Some recent epidemiological studies indicate that BPA exposure may either enhance [58,59] or decrease [55,60] serum T4 levels in humans. These observations are in line with in vitro data showing that BPA is a weak ligand for TRs, therefore acting as an indirect antagonist [61,62], and may also interfere with thyroid hormone action by a nongenomic mechanism [63].

The parathyroid gland has recently been suggested as a target for EDC action [61]. A panel of EDCs has been tested in human parathyroid tumors by metabolomics and mass spectrometry, showing that PCBs, PBDEs, and dichloro-diphenyl-trichloroethane (DDT) derivatives were associated with parathyroid tumor growth, having an agonist hormonal receptor action, and were negatively correlated with patients’ serum calcium [64]. Further studies are needed to confirm the role of EDCs in the parathyroid gland.

Take home message: Much evidence indicates that a variety of natural and industrial chemicals may interfere with thyroid hormone synthesis, transport, metabolism, and clearance acting as receptor agonists or antagonists. Other chemical compounds, such as PCBs and PBDEs can act as hormone modulators. The final effect of exposure to a single EDC or a mixture of them may cause a reduction in thyroid hormone levels. The conflicting results on this aspect may depend on both the predominant effect of the EDC combinations and the partial adaptive response of the hypothalamus–pituitary–thyroid axis. Hence, circulating levels of thyroid hormone may not be assumed as the hallmark of EDC effects on the thyroid system.

## 3. Adrenal Glands

The adrenal gland is highly sensitive to EDCs due to specific biochemical features including high vascularization, high lipophilicity due to fatty acid contents (steroid hormones), and the presence of cytochrome P450 enzymes that produce free radicals and toxic reactive compounds [65]. The human adrenocortical cell line H295R has been used to assay many EDCs because it expresses all the enzymes necessary for steroidogenesis [66].

EDCs induce adrenocortical toxicity by stimulating or inhibiting steroidogenic enzymes such as the steroid acute regulatory protein (StAR), aromatase, 3β-, 11β-, and 17β-hydroxysteroid dehydrogenases [67,68]. EDCs with inhibitory effects on adrenal steroidogenesis include etomidate, which was the first EDC identified, directly inhibiting 11β-hydroxylase and consequently cortisol synthesis [69], as well as mitotane [70], ketoconazole [71], cardiac glycosides [72], nitrofurans [73], and astazene [74]. By contrast, other agents have proved to increase the activity of steroidogenesis enzymes; one of these agents is PCB126, which can stimulate aldosterone biosynthesis and increase expression of the angiotensin 1 (AT1) receptor, enhancing the angiotensin II responsiveness of adrenal cells [75]. Similarly, lead has been reported to increase aldosterone synthesis, upregulating the 11β-hydroxylase 2 [76]. Further, the herbicide (2-chloro-s-triazine herbicides) stimulates expression of CYP19, which encodes aromatase, increasing adrenal estrogen secretion [77].

Recent studies have investigated the effects of the pesticide DDT at high and low doses, proving it to be toxic and disruptive for the glomerulosa and reticularis zones. The zona fasciculata was less damaged by low (supposedly non-toxic) exposure to DDT and its metabolites but affected by toxic levels of exposure [78]. A study on prenatal and postnatal exposure to low doses of DDT showed retarded development of the reticularis zona in treated rats compared to controls, impairing sexual development, due to low expression of β-catenin [79]. Another study showed that prenatal and postnatal exposure to low doses of DDT reduced the development of the adrenal medulla and synthesis of tyrosine hydroxylase, resulting in low epinephrine secretion [80].

Exposure to an antifungal agent, triadimefon, was tested in pregnant female rats. It resulted in inhibition of development of the adrenal cortex in male fetuses secondary to inhibition of synthesis of steroid hormones [81].

Recently, in addition to chemical EDCs, an important role of non-chemical compounds has emerged, such as artificial light at night (ALAN). Chronic exposure to ALAN, even for a short duration, activates the hypothalamus–pituitary–adrenal (HPA) axis, increasing glucocorticoid concentrations, disturbing the circadian rhythm [7]. Currently, there are no reported effects of radiation exposure to the adrenal gland, which appears to be quite resistant to radiation exposure [82].

Other chemical compounds, able to modulate adrenal cell signaling pathways include phytoestrogens and xenoestrogens, which can indirectly impact adrenal steroidogenesis [83].

Take home message: EDCs have a significant impact on adrenal steroidogenesis. They can act as receptor antagonists directly inhibiting 11β-hydroxylase and consequently cortisol synthesis or agonists stimulating aldosterone biosynthesis by regulation of AT1 or 11β-hydroxylase 2 receptors. Further, some of them can stimulate the expression of CYP19, increasing adrenal estrogen secretion. Non-chemical compounds including ALAN and chemical agents such as phytoestrogens and xenoestrogens can act as hormonal modulators.

## 4. Pancreatic Beta-Cells

Very little is known about the pattern of interaction between endocrine disruptors and pancreatic β-cell function. The most important evidence of interaction comes from in vitro studies. According to the latter, xenoestrogens and in particular BPA are shown to affect the signaling system of pancreatic β-cells by binding to the non-classical membrane estrogen receptor at low, nanomolar concentrations [11,84]. An in vivo experiment aimed at evaluating the impact of BPA on pancreatic β-cell function demonstrated that acute administration of BPA in adult male mice induced a rapid change in glycemic balance, with by a decrease in glucose levels and a rise in insulin levels. Moreover, after four-day administration of BPA a significant intracellular content of insulin was found in treated mice compared to untreated ones. When testing the mice for glucose tolerance, peripheral insulin resistance was detected [85,86,87].

The potentiality of EDCs to affect the other important components of pancreatic islets, i.e., alpha-cells, when low doses of BPA were added in cultured intact islets has also been evaluated, which caused an impairment of Ca^2+^ signals exclusively in pancreatic alpha cells. The latter cells were also negatively influenced in their secretion when low glucose levels were reached, demonstrating alteration in the regulation of the crosstalk between alpha and β-cells. These data confirmed that pancreatic cells are target cells for endocrine disruptors, determining a stimulatory effect of insulin secretion and an inhibitory effect on alpha cells, drastically reducing intracellular calcium ion oscillations induced by low glucose levels [12,87].

From these observations it is conceivable that in type 1 diabetes mellitus (T1DM) the autoimmune process affecting β-cells may be initiated by environmental contaminants, such as per- and poly-fluoroalkyl substances (PFAS). In the literature, it has been reported that in children and adolescents PFAS levels were higher at the onset of diabetes than in healthy controls. It has also been hypothesized that prenatal exposure to PFAS causes alteration of the lipid profile in infants, which in itself could increase the risk of insular autoimmunity in T1DM [13,15,16].

However, it is very difficult to demonstrate a direct effect of exposure to EDCs on a disease, especially when a large number of years elapse [11,12]. We can report the case of a teenager who after having practiced a diet only based on fish to lose weight showed alopecia and the typical signs and symptoms of T1DM with high-titer positivity of the autoantibodies versus pancreatic β-cells. We assayed several EDCs in the hair, blood, and urine and found a high amount of mercury in all the samples, but when we tested her family, we also found high values in their samples and yet they did not develop T1DM (personal unpublished data). Another case is that of a mother and daughter who developed T1DM a few days apart after working in a wine cellar. The list of EDCs that could exert a β-cytotoxic effect was very complex, although the negativity of the blood samples for known viruses was confirmed both at the onset and in the following months (personal unpublished data).

Due to the non-persistent nature of EDCs, the consistency of these chemicals has been hypothesized but a wide range of within-person variability was confirmed. It remains unclear how ubiquitous exposure levels change over time and how consistent exposure remains over the course of multiple years [11,19,88,89,90].

Although there are not many up-to-date studies in the literature, the first in vitro experiments have confirmed the hypothesis that EDCs can act directly or indirectly on the pathogenesis of diabetes, which is certainly the largest epidemic worldwide [11,85].

Beta cells are metabolically highly active, providing a defense mechanism which plays a particularly important role in the maintenance of cell function and survival. Studies conducted on rodent β-cell lines evaluated the effects of chemical compounds including streptozotocin, alloxan, ninhydrin, and hydrogen peroxide [88].

Alloxan monohydrate (5,6-dioxyuracil or 2,4,5,6-tetraoxypyri-midine), an oxygenated pyrimidine derivative, has shown to be toxic for β-cells mimicking glucose in structure and showing a low-affinity GLUT2 glucose transporters. Alloxan enters β-cell and it is metabolized in dialuric acid. Dialuric acid is oxidized to form hydrogen peroxide, superoxide radicals, hydroxyl radicals, and an alloxan radical, which is an important toxic intermediate in the redox cycling reactions. Ninhydrin, a stable analog of alloxan is non-diabetogenic in rats. Interestingly, it shows a different toxicity depending on its concentrations. At low concentrations, it is toxic for pancreatic β-cells only, while at high concentrations, it is toxic for all types of endocrine cells and exocrine pancreatic cells [91].

By contrast, streptozotocin (2-deoxy-2-([(methylnitrosoamino) carbonyl] amino)-D-glucopyranose), an antibiotic and antineoplastic agent produced by streptomyces achromogenes, is diabetogenic. It is an alkylating agent. Like alloxan, it enters β-cells via low-affinity GLUT2 glucose transporters and has been shown to be nontoxic to RINm5F cells, which do not express GLUT2. Prolonged exposure to streptozotocin causes an alkylation of mitochondrial DNA and proteins, resulting in a decreased insulin secretion. Thus, streptozotocin causes β-cells apoptosis by excessive DNA damage. A study conducted on human 1.1B4 cells has shown the molecular mechanisms of toxicity and cell damage mediated by streptozotocin, alloxan, ninhydrin, and hydrogen peroxide. In addition, it has shown new horizons for the treatment and possible prevention of β-cell loss in diabetes [91,92].

Several studies have suggested that exposure to chemical agents results in increased degranulation and/or a loss of β-cell identity as supported by the defect in insulin production and the presence of polyhormonal cells in the pancreases of patients with T1DM. These findings tend to support the concept of a “β-cell identity crisis”, where β-cells dedifferentiate into other endocrine cells (α-cells or δ-cells) as a defense mechanism [93]. Along with this β-cell identity crisis, levels of ‘semi’ β-cells that only express chromogranin A (chromogranin-positive, hormone-negative (CPHN) cells) are increased in the pancreas from patients with T1DM and T2DM and they are scattered throughout the pancreas regardless of inflammation level. More recent findings suggest that the primary functions of cytokines and nitric oxide are to protect islet endocrine cells from damage, and only when regulation of cytokine signaling is lost does irreversible damage occur. In support of this hypothesis, large increases in serum IL-1β values have been observed in response to an acute infection or to BPA. Beta-cell islets are highly vascularized and receive the majority of the pancreatic blood flow. This vascular architecture is essential for the control of glucose homeostasis, but also leads to β-cells being bathed in IL-1β during an infection or environmental insult. Furthermore, β-cells have limited capacity to proliferate. Therefore, if cytokines were solely damaging for β-cells, we might expect a higher incidence of T1DM than currently observed. In this light, cytokines may stimulate protective gene expression in β-cells, and, through production of nitric oxide, protect β-cells from apoptosis or viral infection or environmental insult [94].

Take home message: Pancreatic cells are target cells for EDCs, as alloxan compounds and streptozocin, having a stimulatory effect on insulin secretion and an inhibitory effect on alpha cells. Other agents, such as xenoestrogens and BPA can be involved in the pathogenesis of T1DM acting like hormone modulators in the cell signaling pathway.

## 5. Testes

Most EDCs have an estrogenic or anti-androgenic function [95,96,97,98].

The decrease in male semen quality has become a major concern in European countries. Various studies have suggested a 7% decrease in fertility [98] due to reduced hormone concentrations as well as worsening semen parameters [99]. A progressive reduction in testicular volume in association with the worsening of seminal parameters has also been reported. These changes could be related to increasing exposure to environmental contaminants [99]. Many studies have demonstrated a significant worsening in semen parameters in animals exposed to EDCs [100]. As stated before, EDCs classically act on the AR or the ER. Their effects on the male reproductive system are due to interference with production by, or functioning of, steroid hormones, in particular inhibition of 5α-reductase and aromatase activities [96].

Phthalates are widely employed in many commercial products and have antiandrogenic and estrogen-like effects that make them act as EDCs. Some studies have demonstrated a reduction in androgen concentrations [101] and a decline in semen quality following exposure to phthalates [102]. In ex vivo studies, sperm motility was decreased and long-term exposure to phthalates determined cytotoxicity [103]. Studies in human populations have confirmed the reduction in sperm motility [102].

BPA is widely employed in manufacture of consumer products. It binds to ER-α and -β, thereby determining some estrogenic activity [104] that potentially interferes with testicular steroidogenesis; thus, making it a molecular target of BPA. In experimental studies BPA binds AR, the thyroid hormone receptor, and the peroxisome proliferator-activated receptor (PPAR)-γ [105]. BPA seems to be one of the greatest agents of interference with testicular function and may impair either testicular development or testosterone production [98]. In animal models, BPA is not only able to reduce testosterone concentrations but also to decrease sperm counts and motility and to worsen DNA damage [106]. In humans, many studies have investigated the link between urinary BPA levels and semen parameters but the results have been inconclusive [96].

During the last few decades many clinical studies have defined a harmful effect of heavy metals on testicular function. Elevated mercury concentrations have been related to infertility, with alterations of sperm count, motility, and morphology. These effects are caused by detrimental effects on steroid hormone production and directly on Leydig cells [107].

Lead (Pb) may also be involved in alteration of fertility because of a reduction in seminal parameters. Moreover, patients with high Pb seminal concentrations had a lower rate of fecundation with PMA. These alterations are due to multiple complex mechanisms such as DNA fragmentation and many others [107].

Cadmium (Cd) can be found in waters or inhaled through cigarette smoking. It has been suggested it is a potential disruptor of androgen action [108], and besides its proatherogenic effects it may act as a potential chemical reducing testosterone production. It seems that low Cd levels are also related to alterations in spermatic parameters because of Cd accumulation in sperm [109]. Despite this, no studies that relate cadmium and impaired testicular function [109] are present in the literature, so this aspect needs further investigation in humans.

Recently, the incidence of male congenital development defects of the reproductive system (i.e., hypospadias), infertility, and testicular germ cell cancer (TGCC) has been growing, especially in developed countries [97]. Many studies using animal models have confirmed EDC toxic effects, but human studies have been inconclusive in demonstrating a causal link [97]. However, the strongest evidence is that EDC exposure during the prenatal period increases the risk for reproductive disorders [110].

In particular, it seems that phthalates could interfere with normal testicular development in rodent models, but the same effects have not been reported in human studies even though antiandrogenic effects occur. PCBs have been related with urogenital abnormal development in animal models [111]. The results on BPA are controversial [95]. Moreover, it seems that phthalates and BPA are negatively associated to adrenarche and pubarche onset whenever exposure occurs during fetal growth [112].

Other compounds such as perfluorooctanoic acid (PFOA) and perfluoroctanesulfonic acid (PFOS) seem to interfere with the Leydig and Sertoli cell functions, respectively [113,114].

During puberty, many hormonal and clinical changes occur because of hypothalamus–pituitary–gonadal axis activation. Exposure to several compounds could interfere with puberty and thus delay it. Similar effects are exerted by dioxin-like compounds, organochlorine pesticides, and Pb [115]. By contrast, PCBs could bring puberty forward [116]. Finally, fetal exposure to PCBs seems to negatively influence genital maturation [117].

The distance between the anus and the genitalia—anogenital distance (AGD)—could be considered a marker for EDC exposure [118] and a predictor for reproductive disorders in adult life [119].

In murine models, phthalates, PCB, and other compounds with AR antagonist action (i.e., vinclozolin) have been reported to affect male AGD [95] independently from the doses tested [118,119]. Controversial results have been reported in epidemiological studies involving humans [95]. Even though in some studies no correlation between AGD and EDCs were reported, Swan et al. suggested that AGD was inversely related to urinary concentrations of phthalate metabolites in 134 newborns [120]. This result was recently confirmed in 2015 [121].

PFAS, due to their long half-life, could be considered a marker of prenatal exposure despite measurement in newborns. Di Nisio et al. related direct exposure to them with a 10% reduction in AGD and penile length [122]. Moreover, maternal exposure to PCBs was associated with reduced penile length by Guo et al. [123]. However, this result is not univocal: Leijs et al. did not find any correlation between penile length, PCBs, and dioxin-like compound exposure [124].

As regards testicular cancer, evidence suggests that EDC exposure is one of the main reasons for the 2–4-fold increase in the incidence of TGCC during the last 50 years [125]. 1,1-dichloro-2,2-bis- ethylene (p,p’ DDE), a DDT metabolite, and PCBs serum levels have been related to TGCC by some studies, while others have failed to establish a correlation, so the results remain ambiguous [97]. Evidence of the occurrence of cryptorchidism and hypospadias is poor. Some studies have related BPA and phthalic acid to hypospadias [126]. Similarly, pesticide exposure has been related to hypospadias and cryptorchidism by Carbone et al. [127]. By contrast, Main et al. did not relate phthalate monoesters to cryptorchidism [128]. Similarly, PCBs have not been related to cryptorchidism [129]. Despite extensive research on EDC exposure, information about the consequences is still limited and a matter of debate, due to the limited number of human studies. However, animal models have shown the different mechanisms through which EDCs [130,131] could interfere with endocrine systems and how these alterations could be inherited by the next generations.

Take home message: In recent decades, worsening of semen quality related to environmental exposure to different pollutants and interfering agents remains controversial. Although robust evidence on the impact of EDCs on seminal quality in humans is lacking, initial data suggest that chronic exposure to phthalates, BPA, dioxin-like, Pb- and Cd-compounds represent a potential threat for male fertility, due to their interaction with AR or ER and need attention from the andrologist in order to pursue preventive strategies during adolescence and early adulthood.

## 6. Ovaries

EDCs, chemical, and non-chemical hormonal modulators have been reported to affect the female reproductive system.

ALAN is an environmental endocrine modulator that reduces melatonin secretion, increases estrogen/progesterone signaling, and reduces oocyte oxidation [7,132].

Non-ionizing (radio-frequency and extremely low-frequency) electromagnetic field radiation exposure during embryonic development reduces release of estradiol and the ability to reach a developmental stage, an essential prerequisite for reproductive success, and to resume meiotic maturation on in vitro mouse pre-antral follicles, impairing mammalian female reproductive potentiality [133]. Furthermore, it was observed in rats that the nuclei of the oocytes became smaller and changed shape, ovarian follicles underwent atresia and alteration, and differentiation of oocytes and folliculogenesis were affected, resulting in decreased ovarian reserves, leading to infertility or reduced fertility [134].

Ionizing radiation effects, such as in oncological radiotherapy, are mainly local. Pelvic irradiation has a gonadotoxic action with long-term effects leading to ovarian and uterine carcinoma, ovarian insufficiency with follicular atrophy and follicular storage reduction, and pubertal arrest, resulting in permanent or transitory reduction in fertility. The extent of the damage depends on several factors such as younger age, exposure dose and time, and associated chemotherapy, if present [135]. Brain irradiation can reduce pituitary perfusion disrupting the hypothalamus–pituitary–ovary axis, leading to precocious puberty [8].

Stress has been studied as a non-chemical endocrine modulator for the reproductive system in mice, able to activate the hypothalamus–pituitary–adrenal axis [136,137]. The activated hypothalamus–pituitary–adrenal axis suppresses the hypothalamus–pituitary–ovarian axis [138]. Moreover, stress can induce impairment in follicular and oocyte development [139], blockage of ovulation, reduction in steroidogenesis [138], and an irregular estrus cycle [140]. Stress also results in oxidative damage due to increased production of ROS in the follicular fluid environment and significant reduction in the antioxidant enzymes SOD, GST, GPx, 3β-HSDH, and breaks in the DNA in the ovary, resulting in an increase in the percentage of apoptotic granulosa cells and early ovarian senescence [140,141]. Stressed rats showed a significant decrease in the number of small follicles, and serum estradiol, AMH, and GnRH levels, prolonged estrous cycle, and increased serum FSH levels. Furthermore, ovarian stroma revealed severe fibrosis, cortical thickening, and a disorganized structure, while atretic follicles were markedly increased and the corpus luteum exhibited fibrosis and an increase in number [142,143].

Classically, EDCs are the most widely studied of endocrine disruptors. Exposure to BPA can cause female fertility problems related to impairment of folliculogenesis and steroidogenesis, alterations in ovarian morphology, changes in uterine morphology and function with endometriosis, grafting and embryo implantation difficulties [144,145,146,147]. BPA can interfere with steroidogenesis with dose-dependent and species-dependent effects, as described in several articles conducted on sheep, zebrafish, and mice [148,149]. In particular, studies carried out on interstitial cells of rat ovarian theca have shown that exposure to low doses of BPA (0.1–10 µM) increases progesterone levels with increased expression of cytochrome P450 enzymes involved in steroidogenesis. By contrast, exposure to high doses of BPA (1–100 µM) leads to a reduction in aromatase, resulting in reduced levels of estradiol [150].

In addition, at high doses (above 200 µM), BPA determines activation of the aryl hydrocarbon receptor (AHR) signaling pathway with reduced expression of the enzymes involved in synthesis of estradiol and increased expression of the enzymes that metabolize it [151]. Human studies have shown that exposure to BPA, even at low doses, is able to modify the morphology of granulosa cells and reduce the levels of testosterone, progesterone, estrogen, and gonadotropins [152]. Finally, BPA is able to mimic the action of estrogen by bonding competitively with ERs and thus increasing estrogen function. This action mechanism causes a change in expression of ERs and consequently of their target genes, with dysfunction of the female reproductive system [153].

In vitro studies have shown that Di- (2-ethylhexyl) phthalate (DEHP) promotes primordial follicle recruitment and follicle growth via the phosphatidylinositol 3-kinase (PI3K) pathway. This effect is likely to be mediated by its metabolite: mono- (2ethylhexyl) phthalate (MEHP) [154]. Moreover, exposure to DEHP leads to a reduction in estradiol levels, probably through lower expression of Cyp19a1, as confirmed by in vivo (mice) and in vitro studies [155]. The pesticide methoxychlor is able to increase the expression of AMH which, in physiological conditions, acts by recruiting the primordial follicle [156].

Interesting, especially for its transgenerational effects, is 2,3,7,8-Tetrachlorobenzo-p-dioxin (TCDD), which is generated, in a collateral manner, in the processes of pesticide production and waste incineration. In this regard, in 2020, Yu K. et al. showed that exposure of female rats to TCDD caused an increase in serum AMH levels with reduction of primordial follicles in the second generation [157]. Furthermore, TCDD also interferes with the pituitary–ovary axis, modifying the secretion of prolactin (PRL) in the follicular and luteal phases and enhancing luteinizing hormone (LH) secretion in the follicular phase [158].

Phenols induce apoptosis and autophagy of the cells of the rat ovary granulosa; thus, promoting ovarian failure and follicular atresia [159]. Furthermore, Yu PL et al. studied the effect on rats of exposure to low doses of nonylphenol (NP), highlighting an increase in progesterone expression in granulosa cells [160]. Studies carried out on pregnant women have analyzed the effect of NP on the hypothalamus–pituitary–ovary axis, revealing an inversely proportional relationship between urinary NP levels and plasma concentrations of LH [161]. Exposure to high concentrations of NP in the second trimester of pregnancy can also cause small gestational age (SGA) and low neonatal weight and body length at birth [161,162].

Among heavy metals, the best known are Pb, arsenic, and mercury (Hg). In particular, Pb affects the morphology of the ovary and reduces the number of primary follicles, interfering with follicular growth [163,164]. Cadmium (Cd) can cause impaired expression of hypothalamic genes, high concentrations of LH, and low concentrations of AMH in plasma, and abnormal follicular growth with associated reduction of antral follicles. The main results of these effects are alteration of the estrus cycle, premature ovary failure (POF), and polycystic ovary syndrome (PCOS) [165,166].

With regard to Hg, studies have shown that it can increase the incidence of spontaneous abortion, preterm birth, and congenital malformation. Furthermore, in 2019, a study carried out on rats revealed that Hg can also impair the estrous cycle and follicular growth with decreased antral follicles and increased follicular atrophy, as well as increased cystic ovarian follicles [167,168,169].

Another important EDC is diethylstilbestrol (DES), a non-steroidal estrogen used in the past as an anti-abortive drug. It has a transgenerational action as exposure to DES during pregnancy can impair development of the reproductive system, increasing the risk of developing cervical and/or vaginal clear cell adenocarcinoma in daughters [170,171].

Genistein is a phytoestrogen isoflavone that acts by binding to the estradiol receptor and increasing its expression in the ovary. This phytoestrogen determines an impairment of folliculogenesis, decreases the number of follicles (primordial, primary, and secondary), and promotes recruitment of antral follicles [172,173]. Moreover, some studies have revealed a non-monotonous dose-dependent effect of genistein in the production of steroid sex hormones. In this connection, low serum levels of genistein cause an increase in progesterone, while high concentrations of genistein result in its reduction [174,175].

EDC mixtures can damage several metabolic processes, even at low doses when compared to the effects of the single chemical agents [176,177]. They can mimic the normal hormone (LH, FSH, testosterone, and estrone) functions, hence disturbing the normal reproductive process and interfering with female sexual maturity and the estrogen cycle [178].

In vitro exposure of cultured mouse follicles to a mixture of phthalate metabolites (MEP, MEHP, MBP, MiBP, MiNP, and MBzP) upregulated Cyp1a1, an estradiol-metabolizing enzyme, increased expression of Star, while it decreased all downstream enzymes [179].

A mixture of phthalates (MEP, MEHP, MBP, MiBP, MiNP, and MBzP) affects cell cycle regulators, apoptotic factors, and several receptors and receptor-associated genes, inhibiting growth of mouse antral follicles [180]. Likewise, prenatal exposure to a different quinary phthalate mixture (DEP, DEHP, DBP, DNOP, DIBP, and BBP) induced increased uterine weight, cystic ovaries, disrupted the estrous cycle, and reduced fertility in female rats [180].

A combination of microplastics and heavy metals caused significant alteration in the hypothalamic–pituitary–gonadal axis and consequent follicular atresia with empty follicles in female fish [181].

In rats, a mixture of soy isoflavones (genistein + daidzein + glycetin) decreased the primordial and primary follicle number, and increased follicular atresia, large antral follicles, and multiple and binuclear oocytes. Moreover, less estradiol and progesterone and more cholesterol and cortisol are observed in exposed rats [182].

Furthermore, EDC mixtures can have transgenerational phenotype effects, not produced directly by the chemicals to which subjects are exposed [183,184]. Ternary plasticizer mixtures (BPA + DEHP + DBP) aggravated the epigenetic transgenerational inheritance in adult rats. Epimutations, i.e., differential DNA methylation regions in gene promoters, and a significant increase in abnormalities, such as polycystic ovaries, primordial follicle loss, and primary ovarian insufficiency, were observed in subsequent generations [185,186].

Moreover, pharmaceutic mixtures (CBZ + DIC + EE2 + MET and EE2 + NOR) are believed to disturb the androgen/estrogen level balance, reducing the age at first reproduction, limiting reproductive fertility, and hence reproductive success. A reduced number of offspring was observed in crustaceans for up to four generations [183,187].

In female juvenile trout, a pharmaceutic mixture of paracetamol, carbamazepine, diclofenac, irbesartan, and naproxen significantly increased serum 11-ketotestosterone and estradiol, ovarian expression of Esr1, and steroidogenic factors Cyp17a1, Cyp19a1, and Hsd11B [188].

Take home message: Non-chemical agents, such as ALAN, radiation, and stress have an inhibitory effect on estrogen production and signaling, acting as hormone modulators. EDCs can act in a non-monotonous dose-dependent way, stimulating or inhibiting ER. Notably, BPA and ginestein at high doses can have an inhibitory effect on ER and at low doses can have a stimulatory effect on ER. Further, ECD mixtures can have transgenerational effects modifying the phenotype of offspring generations.

## 7. Conclusions

We reviewed current knowledge of EDCs and chemical and non-chemical hormone modulators in the endocrine system. The complex mechanisms of regulation of endocrine disruptors and endocrine function are still only partially investigated. There are many difficulties and limitations in research on EDCs and this may be related to difficulty in translation of what happens in humans. Clear evidence was obtained from animal, biochemical, and human studies that some specific chemicals may affect hormone activity with important consequences for public health. However, the critical question is whether the level of exposure humans encounter is sufficient to produce demonstrable and harmful effects, and this presently remains unclear.

Long-term studies are required to clarify and explain the dose–effect relationship between chemical endocrine disruptors and endocrine disease. EDCs may interfere with the endocrine system at different steps of hormone production, regulation, transport, and action, acting as antagonists or agonists of hormone action (Figure 1).

Several lines of research are ongoing about identification of reliable biomarkers of hormone action in human tissues to be used in epidemiological studies. Finally, given the importance of all hormones in regulating many body functions, there is an urgent need for novel biomarkers, detectors, and assays for early detection of EDC effects.

Non-chemical agents, such as ALAN, radiation, stress, and temperature can impact the endocrine system acting as hormone modulators, while chemical agents can have a disrupting or hormone modulating action.

In the near future, it would be necessary to use specific methods to screen the agonist/antagonist potential of EDCs on endocrine receptors and the hormonal modulation effect of chemical and non-chemical agents, in order to start a targeted therapy.

## Figures and Tables

**Figure 1 ijms-23-05710-f001:**
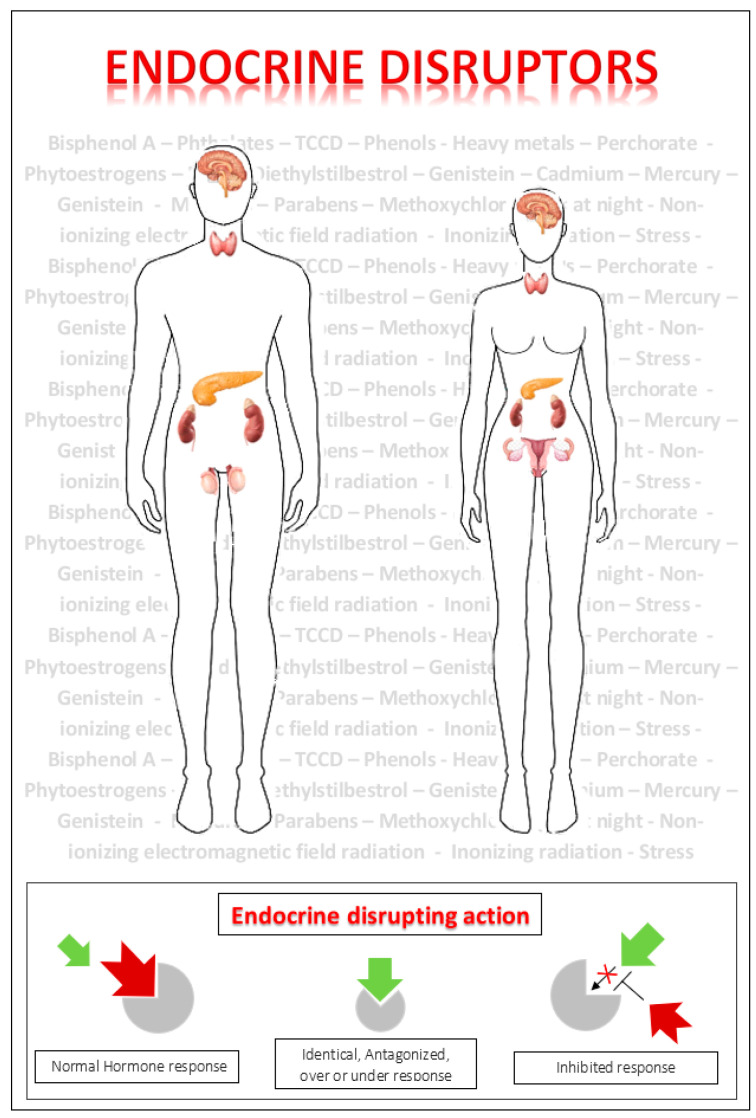
Chemical endocrine disruptors interfering with the endocrine system mimicking, antagonizing, under- or over-stimulating hormonal response.

## Data Availability

Not applicable.

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
