# Peer review of "Impact of Chemical Endocrine Disruptors and Hormone Modulators on the Endocrine System"

_ijms, 2022, doi:10.3390/ijms23105710_

Round 1

Reviewer 1 Report

Review of manuscript entitled „Impact of chemical and Non-chemical endocrine disruptors in the endocrine system”m authored by Guarnotta et al. (Manuscript ID: ijms-1690847).

Today, human societies live in a harsh environment filled with countless chemicals and non-chemical compounds: a living milieu that had been substancially cleaner a few generations ago, and that has been polluted to dangerously high levels by now. Research on the effects of environmental pollutants on human and animal health is accelerating and pours a miriad of data to be analyzed and interpreted. Therefore, review articles in this topic are of increased value, as they are expected to provide a qualitiy interpretation of research data that is based on a didactic basis.

While the present review manuscript collected important and interesting pieces of scientific information, the manuscript itself remains just that: a collection of data, without the didactic approach and without a thoughtful interpretation, without a meaningful take-home message to the reader.

The very first and major problem with the manuscrpt is that it does not make a distinction between endocrine disrupting chemicals (endogenous or exogenous chemicals with the potency to mimic or antagonize hormone effects, i.e., hormone agonists or antagonists) in a classical sense, and chemical and nonchemcal compounds with the potency to modulate the function of endocrine glands (hormone system modulators). In this respect, in the authors’ view the scalpel of a surgeon that is used to remove an endocrine gland would also be classified as an endocrine disruptor. Pouring all collected information into this highly mixed bowl does not help in determining our way out of the current chemically infiltrated world.

The second problem is the lack of the added valued of a meaningful scientific interpretation. Like it was mentioned above, the manuscript is a pure collection of (otherwise interesting) data, thrown together without respect to any didactic approach that has been folloved so far in the relevant literature. Inclusion of the effects of electromagnetic fileds into this collection may be a good idea, however, the authors should realize that with this they touch an intensely growing filed of research and, therefore, picking only the few sentences from that pool of data may not be worthy of the subject chosen be the authors.

Altogether, this reviewer does not recommend the publication of this manuscript in the present from, however, taking a more thoughtful approach could trnasform the presented data collection into a valuable review article.  

Author Response

Reviewer 1

Review of manuscript entitled „Impact of chemical and Non-chemical endocrine disruptors in the endocrine system”m authored by Guarnotta et al. (Manuscript ID: ijms-1690847).

Today, human societies live in a harsh environment filled with countless chemicals and non-chemical compounds: a living milieu that had been substancially cleaner a few generations ago, and that has been polluted to dangerously high levels by now. Research on the effects of environmental pollutants on human and animal health is accelerating and pours a miriad of data to be analyzed and interpreted. Therefore, review articles in this topic are of increased value, as they are expected to provide a qualitiy interpretation of research data that is based on a didactic basis.

While the present review manuscript collected important and interesting pieces of scientific information, the manuscript itself remains just that: a collection of data, without the didactic approach and without a thoughtful interpretation, without a meaningful take-home message to the reader.

The very first and major problem with the manuscrpt is that it does not make a distinction between endocrine disrupting chemicals (endogenous or exogenous chemicals with the potency to mimic or antagonize hormone effects, i.e., hormone agonists or antagonists) in a classical sense, and chemical and nonchemcal compounds with the potency to modulate the function of endocrine glands (hormone system modulators). In this respect, in the authors’ view the scalpel of a surgeon that is used to remove an endocrine gland would also be classified as an endocrine disruptor. Pouring all collected information into this highly mixed bowl does not help in determining our way out of the current chemically infiltrated world.

The second problem is the lack of the added valued of a meaningful scientific interpretation. Like it was mentioned above, the manuscript is a pure collection of (otherwise interesting) data, thrown together without respect to any didactic approach that has been folloved so far in the relevant literature. Inclusion of the effects of electromagnetic fileds into this collection may be a good idea, however, the authors should realize that with this they touch an intensely growing filed of research and, therefore, picking only the few sentences from that pool of data may not be worthy of the subject chosen be the authors.

Altogether, this reviewer does not recommend the publication of this manuscript in the present from, however, taking a more thoughtful approach could trnasform the presented data collection into a valuable review article.  

Thanks for the interesting comments. We modified the text, adding a take home message at the end of each paragraph trying to provide information about the effects (agonist or antagonist) of EDCs.

Reviewer 2 Report

General comments:

- The clarity of English language use throughout the manuscript needs to be improved to ensure that readers can accurately assess and understand the work that was conducted. I have provided some line-by-line feedback on this issue (see below) but stopped after the Abstract to concentrate on scientific details. However, the clarity of language must be improved throughout the entire document.

- The organisation of the sections and subsections is potentially confusing as some are far longer than others, and some subsections within the same section have quite a different focus / scope. For example, within the section ‘Thyroid’, subsection topics vary considerably, from discussing effects that EDCs can have on the thyroid, to specific chemicals that can impact the thyroid. In my opinion, some attention should be paid to potentially grouping some of the article’s subsections so that they are more consistent in length and scope.

Point-by-point feedback:

- Title: Replace ‘in’ with ‘on’.

- Line 16: Replace ‘represented by’ with ‘including’

- Line 17: ‘Radiations’ should be replaced with ‘radiation’.

- Line 18: This should read ‘…can seriously affect the endocrine system.’

- Line 19: Replace ‘in hormonal activity’ with ‘with hormonal activity’.

- Line 21: Replace ‘do not allow to establish a causal relationship’ with ‘do not allow the establishment of causal relationships’

Line 23: ‘Non’ does not need to be capitalised here.

Line 23: Should be ‘the endocrine system’

Line 23: Should be ‘the thyroid’

Line 24: I think that this is unnecessarily complicated language: ‘and their putative interference with hormonal milieu.’ Is there a way to say this more simply and clearly?

Line 36: Which drugs? It might be good to name a few in parentheses here.

Line 41: Changing back and forth between the terms ‘EDCs’ and ‘EDs’ is confusing and potentially unnecessary. As just one example, you say here that there are 400 million EDs, but I think you mean EDCs. I would change this terminology throughout to ‘endocrine-disrupting chemicals (EDCs)’ and ‘non-chemical endocrine disruptors’. Or, at least, be careful where you use each of your acronyms.

Line 46: Again, it’s not clear whether you actually mean EDs or EDCs.

Line 53: Also, behavioural abnormalities.

Line 62: What does ‘side- bars’ mean here? Delete?

Lines 69–70: It would be useful here to include a few sentences describing how the review is organised. I.e. how are the sections structured? Why are they ordered this way? This will help the reader digest the information more easily. This would be especially useful because you have some subsections that are quite long and others that are very short. This may be confusing to readers and so it would be good to justify why the review is organised as it is.

Line 81: Such as? Maybe you could provide a few examples in parentheses here?

Line 115: This section (‘Thyroid’) has many subsections and I’m not sure that all of them are necessary, or at least I’m not sure that all need to be separate subsections because some of the subsection topics overlap. I recommend that you see if any of these subsections can be grouped, to make the paper more digestible for readers.

Line 145: The previous subsections each talk about specific effects of EDCs on the thyroid, whereas the subsections from here forward present classes of chemicals that can disrupt the thyroid. It is potentially confusing to readers to present the subsections this way with no justification for the change between subsection type. I would consider organising the subsections so that there are fewer overall, and so that they are more consistent in topic and scope.

Line 175: This section is extremely short. Does it really warrant its own section or could it be merged with another section? Or perhaps you could add some additional information to this section to make it less sparse.

Line 215: It is my understanding that the accepted acronym for Artificial Light at Night is ALAN, not LAN. In my opinion, the acronym ALAN should be used throughout.

Lines 320–322: This information seems like it would be better suited in the Introduction section?

Line 446: Again, this is confusing because you use both of the acronyms ‘EDC’ and ‘ED’. Here, surely chemical EDs are EDCs? I think this use of terminology could be made clearer throughout.

Line 522: ‘Diethylstilbestrol’ does not need to be capitalised here.

Line 571: I don’t find that the Conclusions section is very forward-looking in discussing where the field of EDCs research is heading. What are the major outstanding questions? What do we need to do to answer them? Is increased interdisciplinary collaboration needed? I feel that, because the rest of the review is recapping work that has already been done, it would be good to include here in the Conclusions section some more forward-looking perspectives on the future of research in this field and why it is so important that we reveal the true impacts of EDCs on human health and the environment.

Author Response

Reviewer 2

- The clarity of English language use throughout the manuscript needs to be improved to ensure that readers can accurately assess and understand the work that was conducted. I have provided some line-by-line feedback on this issue (see below) but stopped after the Abstract to concentrate on scientific details. However, the clarity of language must be improved throughout the entire document.

 Thanks for the comment. An English mother tongue revised the manuscript.

- The organisation of the sections and subsections is potentially confusing as some are far longer than others, and some subsections within the same section have quite a different focus / scope. For example, within the section ‘Thyroid’, subsection topics vary considerably, from discussing effects that EDCs can have on the thyroid, to specific chemicals that can impact the thyroid. In my opinion, some attention should be paid to potentially grouping some of the article’s subsections so that they are more consistent in length and scope.

 Thanks for the comment. We deleted the subsections in the thyroid section.

Point-by-point feedback:

- Title: Replace ‘in’ with ‘on’.

Thanks for your suggestion. We changed it.

- Line 16: Replace ‘represented by’ with ‘including’

 Thanks for your comment. We changed it as you suggested.

- Line 17: ‘Radiations’ should be replaced with ‘radiation’.

 Thanks for the suggestion. We changed it.

- Line 18: This should read ‘…can seriously affect the endocrine system.’

  Thanks for the suggestion. We changed it.

- Line 19: Replace ‘in hormonal activity’ with ‘with hormonal activity’.

 Thanks for the suggestion. We changed it.

- Line 21: Replace ‘do not allow to establish a causal relationship’ with ‘do not allow the establishment of causal relationships’

  Thanks for the suggestion. We changed it.

Line 23: ‘Non’ does not need to be capitalised here.

  Thanks for the suggestion. We changed it.

Line 23: Should be ‘the endocrine system’

Thanks for the suggestion. We changed it.

Line 23: Should be ‘the thyroid’

Thanks for the suggestion. We changed it.

Line 24: I think that this is unnecessarily complicated language: ‘and their putative interference with hormonal milieu.’ Is there a way to say this more simply and clearly?

 Thanks for your comment. We modified it.

Line 36: Which drugs? It might be good to name a few in parentheses here.

 Thanks for your comment. We added it some drugs involved.

Line 41: Changing back and forth between the terms ‘EDCs’ and ‘EDs’ is confusing and potentially unnecessary. As just one example, you say here that there are 400 million EDs, but I think you mean EDCs. I would change this terminology throughout to ‘endocrine-disrupting chemicals (EDCs)’ and ‘non-chemical endocrine disruptors’. Or, at least, be careful where you use each of your acronyms.

 Thanks for your comment. We changed the terminology and used EDCs and non chemical endocrine disruptors.

Line 46: Again, it’s not clear whether you actually mean EDs or EDCs.

 Thanks for your comment. We modified it.

Line 53: Also, behavioural abnormalities.

 Thanks for your comment. We added it as you suggested.

Line 62: What does ‘side- bars’ mean here? Delete?

Thanks for the suggestion. We deleted it.

Lines 69–70: It would be useful here to include a few sentences describing how the review is organised. I.e. how are the sections structured? Why are they ordered this way? This will help the reader digest the information more easily. This would be especially useful because you have some subsections that are quite long and others that are very short. This may be confusing to readers and so it would be good to justify why the review is organised as it is.

Thanks for the comment. We reorganized the text deleting the subsections and making it more easy to read.

Line 81: Such as? Maybe you could provide a few examples in parentheses here?

Thanks for the question. We added some examples in the text.

Line 115: This section (‘Thyroid’) has many subsections and I’m not sure that all of them are necessary, or at least I’m not sure that all need to be separate subsections because some of the subsection topics overlap. I recommend that you see if any of these subsections can be grouped, to make the paper more digestible for readers.

Thanks for the comment. We deleted the subsections and grouped them.

Line 145: The previous subsections each talk about specific effects of EDCs on the thyroid, whereas the subsections from here forward present classes of chemicals that can disrupt the thyroid. It is potentially confusing to readers to present the subsections this way with no justification for the change between subsection type. I would consider organising the subsections so that there are fewer overall, and so that they are more consistent in topic and scope.

 Thanks for the comment. The subsections have been deleted and the text has been better reorganized.

Line 175: This section is extremely short. Does it really warrant its own section or could it be merged with another section? Or perhaps you could add some additional information to this section to make it less sparse.

 Thanks for the comment. Unfortunately, there are no further information about parathyroid glands. We added it in the thyroid section.

Line 215: It is my understanding that the accepted acronym for Artificial Light at Night is ALAN, not LAN. In my opinion, the acronym ALAN should be used throughout.

 Thanks for the suggestion. We changed it.

Lines 320–322: This information seems like it would be better suited in the Introduction section?

 Thanks for the comment. We changed it.

Line 446: Again, this is confusing because you use both of the acronyms ‘EDC’ and ‘ED’. Here, surely chemical EDs are EDCs? I think this use of terminology could be made clearer throughout.

 Thanks for the comment. We used EDCs and non -chemical endocrine disruptors to avoid confusion.

Line 522: ‘Diethylstilbestrol’ does not need to be capitalised here.

 Thanks for the suggestion. We changed it.

Line 571: I don’t find that the Conclusions section is very forward-looking in discussing where the field of EDCs research is heading. What are the major outstanding questions? What do we need to do to answer them? Is increased interdisciplinary collaboration needed? I feel that, because the rest of the review is recapping work that has already been done, it would be good to include here in the Conclusions section some more forward-looking perspectives on the future of research in this field and why it is so important that we reveal the true impacts of EDCs on human health and the environment.

Thanks for this interesting comment. We improved the conclusion following your suggestion.

Round 2

Reviewer 1 Report

The manuscript has been improved, however, the initial conceptual problem of this reviewer had not been taken into account: “The very first and major problem with the manuscrpt is that it does not make a distinction between endocrine disrupting chemicals (endogenous or exogenous chemicals with the potency to mimic or antagonize hormone effects, i.e., hormone agonists or antagonists) in a classical sense, and chemical and nonchemcal compounds with the potency to modulate the function of endocrine glands (hormone system modulators). In this respect, in the authors’ view the scalpel of a surgeon that is used to remove an endocrine gland would also be classified as an endocrine disruptor. Pouring all collected information into this highly mixed bowl does not help in determining our way out of the current chemically infiltrated world.”

Once again: the homeostasis of an organism is controlled and maintained by the neuroendocrine system. Anything that moves any of the homeostatic parameters out of the physiological spectrum is, at the same time, an initiator of changes in the endocrine system. Such factors may be materials as simple and common as the drinking water, or anything that is present in surplus or in insufficient amounts in the metabolic milieu of the organism. Therefore, a clear distinction should be made between real endocrine disruptors and the modulators of endocrine functions. The authors have failed to consider this during their revision of the manuscript.

To highlight this problem a little better:

EDs are of increasing importance in chemical regulations in the European Union, and criteria to identify EDs have recently been presented for two pieces of EU legislation (Biocidal Product Regulation and Plant Protection Products Regulation). The European Commission, in consultation with relevant regulatory bodies, stated that research should improve and harmonize screening and testing protocols/strategies and hazard/risk assessments by developing better and faster tools, test methods or models, including in vitro and in vivo tests, high-throughput and in silico methods (e.g. QSAR), potentially combined with research on adverse outcomes pathways (Call title: New testing and screening methods to identify endocrine disrupting chemicals. Call ID: SC1-BHC-27-2018). In this effort of the EC, currently there are more than 4000 chemical substances with known ability to alter the neuroendocrine regulation of the homeostasis. This is a lot of chemicals, but still in a manageable range, that may be handled in the frames of the above-mentioned EC actions. In contrast, the authors mention about 400 million different chemicals as endocrine disrupting chemicals. This is far more than what could be handled by the legislators and not even true by definition (of what is considered as an endocrine disruptor).

With this comment this reviewer does not recommend the publication of the manuscript in its present revised form.

Author Response

The manuscript has been improved, however, the initial conceptual problem of this reviewer had not been taken into account: “The very first and major problem with the manuscrpt is that it does not make a distinction between endocrine disrupting chemicals (endogenous or exogenous chemicals with the potency to mimic or antagonize hormone effects, i.e., hormone agonists or antagonists) in a classical sense, and chemical and nonchemcal compounds with the potency to modulate the function of endocrine glands (hormone system modulators). In this respect, in the authors’ view the scalpel of a surgeon that is used to remove an endocrine gland would also be classified as an endocrine disruptor. Pouring all collected information into this highly mixed bowl does not help in determining our way out of the current chemically infiltrated world.”

Once again: the homeostasis of an organism is controlled and maintained by the neuroendocrine system. Anything that moves any of the homeostatic parameters out of the physiological spectrum is, at the same time, an initiator of changes in the endocrine system. Such factors may be materials as simple and common as the drinking water, or anything that is present in surplus or in insufficient amounts in the metabolic milieu of the organism. Therefore, a clear distinction should be made between real endocrine disruptors and the modulators of endocrine functions. The authors have failed to consider this during their revision of the manuscript.

To highlight this problem a little better:

EDs are of increasing importance in chemical regulations in the European Union, and criteria to identify EDs have recently been presented for two pieces of EU legislation (Biocidal Product Regulation and Plant Protection Products Regulation). The European Commission, in consultation with relevant regulatory bodies, stated that research should improve and harmonize screening and testing protocols/strategies and hazard/risk assessments by developing better and faster tools, test methods or models, including in vitro and in vivo tests, high-throughput and in silico methods (e.g. QSAR), potentially combined with research on adverse outcomes pathways (Call title: New testing and screening methods to identify endocrine disrupting chemicals. Call ID: SC1-BHC-27-2018). In this effort of the EC, currently there are more than 4000 chemical substances with known ability to alter the neuroendocrine regulation of the homeostasis. This is a lot of chemicals, but still in a manageable range, that may be handled in the frames of the above-mentioned EC actions. In contrast, the authors mention about 400 million different chemicals as endocrine disrupting chemicals. This is far more than what could be handled by the legislators and not even true by definition (of what is considered as an endocrine disruptor).

Thanks for this interesting comment. As you suggested, we tried to delineate a difference in the text between chemical endocrine disruptors and chemical and non-chemical agents acting on hormone modulation. In this light, we also changed the title. We hope that the current changes would be appreciated by the reviewer. Anyway, we are ready to further modify the text, according to reviewer suggestions.  

Reviewer 2 Report

The authors have adequately addressed my concerns in the first round of revision.

Author Response

Thanks for your revision.

Round 3

Reviewer 1 Report

The manuscript is now ready for publication. This revewer recommends the publication of the manuscript.